# Pharmacovigilance in Pediatric Patients with Epilepsy Using Antiepileptic Drugs

**DOI:** 10.3390/ijerph19084509

**Published:** 2022-04-08

**Authors:** Dorota Kopciuch, Krzysztof Kus, Jędrzej Fliciński, Barbara Steinborn, Anna Winczewska-Wiktor, Anna Paczkowska, Tomasz Zaprutko, Piotr Ratajczak, Elżbieta Nowakowska

**Affiliations:** 1Department of Pharmacoeconomics and Social Pharmacy, Karol Marcinkowski University of Medical Sciences, Rokietnicka 7, 60-806 Poznan, Poland; kkus@ump.edu.pl (K.K.); aniapaczkowska@ump.edu.pl (A.P.); tomekzaprutko@ump.edu.pl (T.Z.); p_ratajczak@ump.edu.pl (P.R.); 2Department of Developmental Neurology, Karol Marcinkowski University of Medical Sciences, Przybyszewskiego 49, 60-355 Poznan, Poland; flicinski@hotmail.com (J.F.); bstein@ump.edu.pl (B.S.); awwiktor@ump.edu.pl (A.W.-W.); 3Department of Pharmacology and Toxicology, Institute of Health Sciences, Collegium Medicum, University of Zielona Gora, Licealna 9 Street, 65-417 Zielona Góra, Poland; elapharm@ump.edu.pl

**Keywords:** antiepileptic drugs, pharmacovigilance, epilepsy, adverse drug reactions, adverse events, pediatric population, children

## Abstract

Objective: To investigate the occurrence of adverse effects of antiepileptic drugs (AEDs) in pediatric epileptic patients on mono- or polytherapy. Method: We evaluated eighty consecutive patients that met the following inclusion criteria: aged ≤18 years; diagnosed with epilepsy for at least one year; a stable dose of AED for at least three months; verbal consent to participation in the study. Patients were asked if they had experienced any adverse drug reaction (ADR) related to the AED. Afterward, regardless of the answer, they were interviewed based on a detailed semi-structured questionnaire about the presence of ADRs associated with the AED. The data were analyzed regarding the use of monotherapy or polytherapy. Results: Ninety-seven percent of the patients reported having experienced ADRs related to AEDs. The greatest number of seizures affected the group of patients treated with monotherapy (both at baseline and at followup), but the greatest number of ADRs were observed among patients treated with polytherapy. In patients on monotherapy, the most frequent ADRs reported at baseline included fatigue and somnolence, and among patients with polytherapy, it was fatigue and hair loss. Conclusion: Children on polytherapy were significantly more likely to develop ADRs compared to those on monotherapy, but a statistically significant improvement in seizure frequency was also observed in the group of patients on polytherapy. Pharmacovigilance is very important in children with AEDs, so that ADRs can be identified early and managed appropriately.

## 1. Introduction

Epilepsy is a chronic disorder characterized by episodic gratuitous seizures. Many individuals with epilepsy have a combination of various types of seizures and may have other signs of neurological complications as well [1]. Most patients with epilepsy rely on medical treatment with antiepileptic drugs (AEDs) to achieve control of their seizures [2].

The overall aim in the treatment of epilepsy should be complete control of seizures and no adverse reaction due to medication. In particular, treatment of pediatric epilepsy requires a careful evaluation of the safety and tolerability profile of AEDs to avoid or minimize adverse events (AEs) on various organs, hematological parameters, growth, and pubertal, motor, cognitive, and behavioral development as much as possible [3,4,5,6,7].

Polytherapy is sometimes used in refractory epilepsy despite a significant increase in the number of the side effects [8,9].

Adverse effects of AEDs are typical and can have a significant effect on the quality of life of the patients and add up to treatment letdown in about 40% of admitted patients. The adverse effect summaries of AEDs vary prominently and are often a decisive aspect of drug choice because of the similar efficacy proportions presented by most AEDs. The most communal adverse drug reactions (ADRs) are dose-related and reversible [10].

The aim of this study was to investigate the occurrence of ADRs of AEDs in pediatric epileptic patients on mono- or polytherapy.

## 2. Method

The study was conducted at the department of developmental neurology in Poland, between 2019 and 2020. We evaluated consecutive patients that met the following inclusion criteria: aged ≤18 years, diagnosis of epilepsy for at least one year, a stable dose of AED for at least three months, and verbal consent to participation in the study.

Patients (or/and their caregivers) were asked if they had any ADRs related to the AEDs. After that, regardless of the answer, they were interviewed based on a detailed questionnaire about the presence of ADRs associated with the AEDs (Appendix A). We also assessed specifically the ADRs in the last 3 months.

Patients were interviewed twice during the study, at baseline and followup period.

“Baseline” describes the average number of ADRs per month at the beginning of the study. “Followup” means the average number of ADRs per month over the last 3 months after beginning of the study.

In order to assess whether and how AEDs impact the incidence of ADRs in pediatric patients with epilepsy, the patients’ case histories were analyzed in terms of pharmacotherapy used. Then, the patients were divided into groups as per their treatment, monotherapy or polytherapy. The average number of seizures was calculated for each group. Comparisons of efficacy for treatment regimens (monotherapy vs. polytherapy) were calculated as within-patient ratios of the average seizure frequency at followup, divided by the average seizure frequency at baseline in each group.

To make our results comparable with the metric usually reported in AED clinical trials, we reported the percentage reduction in normalized seizure frequency as (1-SFR) × 100 [11].

## 3. Statistical Analysis

The seizure frequency ratio (SFR) data were obtained after log-transformation of the data and are expressed as means (average) ± standard deviation (SD) and median with 95% confidence intervals (CIs). We used log-transformation of SFR statistics in order to provide a metric symmetrical around SFR = 1 (representing no change); that is, so that small SFRs, reflecting a highly effective trial, would be equally weighted against highly ineffective trials where SFR was high [11].

The data distribution pattern was not normal (unlike a Gaussian function). Statistical analyses for SFR were carried out using the nonparametric Wilcoxon test for paired data. Significant differences between % of group results were determined by analysis of the Test for Proportions.

The study was approved by the Bioethics Committee of the Poznań University of Medical Sciences.

## 4. Results

Eighty epileptic children subjects were included in the study. In total, 53.75% experienced generalized epileptic seizures and 20.00% focal epileptic seizures (Table 1). The age ranged from 2 to ≤18 years. The average duration of epilepsy in study group subjects was 4.11 ± 1.22 years. The children were taking on average 1.79 ± 0.80 AEDs (Table 1).

The largest percentage of children with epilepsy were on levitracetam (LEV) (30.32%) followed by lamotrigine (LTG) (18.88%) and vigabatrin (VGB) (17.68%). The most frequent second generation of AEDs were valproate (VPA) (39.21%) and phentynoin (PHT) (23.32%) (Table 1). Ninety-seven percent of the patients reported having experienced an ADR related to AEDs (*p* < 0.001). In patients on monotherapy, the most frequent ADRs reported at baseline included fatigue (47.09%), and somnolence (55.88%) (Table 2); and at followup, it was emotional liability (50.00%), fatigue (55.88%), psychomotor agitation (61.76%), anxiety (47.05%), and somnolence (58.82%) (Table 2).

In patients on polytherapy, the most frequent ADRs reported at baseline included fatigue (41.30%) and hair loss (34.78%) (Table 2), and at followup, they reported fatigue (58.69%), psychomotor agitation (50.00%), anxiety (50.00%), emotional liability (47.82%), and memory impairment (52.17%) (Table 2).

Statistically significant differences in the incidence of individual ADRs occurred between baseline and followup both in patients on monotherapy and those on polytherapy (Table 2). In both groups, incidence of such ADRs as emotional liability, psychomotor agitation, aggressiveness, and anxiety increased during the followup period. Moreover, in the monotherapy group, tremor and weight gain were much more frequently reported during the followup than during the baseline period. Patients on polytherapy, on the other hand, reported headache, dyspepsia, memory impairment, and lack of focus much more frequently in the followup period than during the baseline period (Table 2).

A statistically significant difference in incidence of hair loss (*p* = 0.0187) and diplopia or blurred vision (*p* = 0.0505) was observed at baseline in the group of patients on polytherapy compared to those on monotherapy. Conversely, in the followup period, statistically significant differences were found for dyspepsia (*p* = 0.0531) and lack of focus (*p* = 0.0003) in the group of patients on polytherapy compared to those on monotherapy (Table 2).

In the followup period, statistically significant differences in clinical data between patients on monotherapy and polytherapy were observed in virtually each of the study clinical parameters, i.e., in the average number of ADRs per month (*p* ˂ 0.0001), average seizure frequency per month (*p* ˂ 0.0001), average number of hospitalization days per month (*p* = 0.0005), and average number of outpatient neurologist visits per month (*p* = 0.0002) (Table 3).

In the baseline period, statistically significant differences in clinical data between patients on monotherapy and polytherapy were not observed only in terms of the average number of AEs per month (*p* = 0.7536) (Table 3).

Similar observations were made in the analysis of occurrence of statistically significant differences in clinical data between the measurements made at baseline and at followup in each of the groups (Table 2). The noted differences were not observed only in terms of the average number of AEs per month in patients on monotherapy (Table 2).

The analysis of average seizure frequency rates (SFRs) depending on the pharmacotherapy regimen (mono- vs. polytherapy) has shown that, both at baseline and at followup, the greatest number of seizures affected the group of patients treated with monotherapy (32.22 ± 12.21 at baseline; 27.47 ± 12.45 at followup) (Table 4). The average SF in patients treated with polytherapy was estimated at 21.12 ± 7.31 at baseline and 15.98 ± 5.19 at followup (Table 4).

A statistically significant improvement in seizure frequency was observed in the group of patients on polytherapy (Table 4). In the group of patients on monotherapy, average seizure frequency was reduced by 15% compared to seizure frequency at the baseline, while in the group on polytherapy, seizure frequency reduction compared to the baseline amounted to 24% (Table 4).

## 5. Discussion

About one-third of patients receiving AEDs in this study developed at least one ADR during the treatment. Similar AED prevalence pattern has been reported in other studies [12,13,14,15].

The number of patients on polytherapy in our study was higher than what was reported in previous studies in the UK [14] and in India [16]. This may result from the fact that many cases of epilepsy in our society are associated with other neurologic disorders, and this may contribute to the difficulty in controlling the seizures with a single AED. Monotherapy is viewed as the initial and preferred option for treating epilepsy, the choice of the drug depends on seizure type and the drug’s efficacy balanced against possible side-effects [17].

In a study conducted in the UK [14], most of the children received monotherapy, with only 25% receiving polytherapy. Various studies suggest that AED used as monotherapy is effective in 60–70% of children [18,19]. Additional drugs in refractory patients have been shown to be only marginally beneficial [20]. Polytherapy entails a greater risk of drug toxicity in pediatric patients in general [21], especially those receiving AEDs [22]. More children receiving polytherapy in this study developed ADRs, with up to a threefold higher incidence of ADRs compared to monotherapy. This is consistent with earlier reports [14,16,22].

Unfortunately, most new AEDs are tested by the pharmaceutical companies as an add-on therapy, and drug toxicity is poorly described. This encourages clinicians to use polytherapy in epilepsy.

A U.S. study of 314 adults found that 44% of the patients were on monotherapy with the remaining 56% of patients on polytherapy [23]. Similar proportions were found in a European study assessing the quality of life of over 5000 patients: 47% of patients were reported to be receiving monotherapy, and 36% were taking 2 AEDs (12% were on 3, 1% on 4 or more, and 4% were not receiving any medication). The drugs most commonly taken were carbamazepine (53%), sodium valproate (33%), and phenytoin (25%) [24].

Although ADRs were more frequent in patients on polytherapy, improvement of clinical parameters such as lower seizure frequency, fewer outpatient neurologist visits, or fewer hospitalization days was observed in this group. The question remains, however, whether the polytherapy may be responsible for this reduction in, for example, seizure control; it is also possible that the second drug alone might be effective. Side effects increase with polytherapy; it is unclear, however, whether this is caused by the number of medications or the total drug load.

The main ADRs identified in this study were behavioral problems such as emotional liability, anxiety, fatigue, and psychomotor agitation, which were comparable to those from another study [25].

In our study the number of adverse effects was similar in both the mono- and polytherapy group. Individual analysis of each side effect, diplopia, anxiety, dyspepsia, dizziness, memory impairment, sleep disturbance, and lack of focus, showed that they were found to occur more frequently in patients on polytherapy.

Similarly to our study results, several other studies have shown that the incidence of adverse effects increases with the number of drugs [26].

Many studies show that comorbidity in epilepsy is a major issue currently, and depression and related symptoms, such as fatigue and lack of focus, are some of the main conditions associated with epilepsy [27,28,29].

Furthermore, the adverse effects of AEDs and emotional/behavioral problems may have the strongest negative influence on the patient’s perception of their current health. The adverse impact of emotional problems on the quality of life of epileptic patients requires an investigation of their presence in every pediatric patient with epilepsy [30].

## 6. Limitation

We are aware that our study had several limitations. Firstly, our study was an observational study and not a randomized controlled trial; therefore, selection bias could have affected the results. The insufficient number of patients recruited may be another limitation. Much larger studies are required to adequately determine the ADRs of AEDs in mono- or polytherapy.

## 7. Conclusions

To conclude, children on polytherapy were significantly more likely to develop ADRs compared to those on monotherapy. Physicians should give AED polytherapy only when the maximum therapeutic doses of monotherapy are ineffective. Pharmacovigilance is very important in children on AEDs, so that ADR can be identified early and managed appropriately. Both clinicians and parents should monitor AED-treated children for adverse reactions, especially for behavioral problems such as emotional liability and anxiety, and fatigue, somnolence, and psychomotor agitation.

## Figures and Tables

**Table 1 ijerph-19-04509-t001:** Demographic and clinical data (at baseline).

**Age; average ± SD**	8.33 ± 4.37
**Gender; *n* (M/F)**	38/42
**Duration of epilepsy, years (average ± SD)**	4.11 ± 1.22
**Type of seizures; *n* (%)**	
Generalized	43 (53,75%)
Focal	16 (20,00%)
Other	21 (26,25%)
**Therapeutic scheme**	
**Monotherapy; *n* (%):**	34 (42.50)
Valproate (%)	39.21%
Phenytoin (%)	23.31%
Carbamazepine (%)	19.80%
Clobazam (%)	10.12%
Levetiracetam (%)	30.32%
Vigabatrin (%)	17.68%
Oxcarbazepine (%)	13.27%
Topiramate (%)	11.40%
Lacosamide (%)	2.63%
Lamotrigine (%)	18.88%
**Polytherapy; *n* (%)**	46 (56.50)
2 drugs (%)	61.94
3 drugs (%)	38.06
**Average AEDs (average ± SD):**	1.79 ± 0.80
Monotherapy	32.22 ± 12.21
Polytherapy	21.12 ± 7.31

SD—standard deviation; AED—antiepileptic drugs.

**Table 2 ijerph-19-04509-t002:** Frequency of adverse events in the last three months versus since the beginning of the study according with treatment groups.

Adverse Drug Reactions	Monotherapy (*n* = 34)			Polytherapy (*n* = 46)			*p* Value	
	Baseline *n* (%)	Followup *n* (%)	*p* Value Baseline vs. Followup	Baseline *n* (%)	Followup *n* (%)	*p* Value Baseline vs. Followup	Monotherapy vs. Polytherapy	
							Baseline	Followup
Emotional liability	3 (8.82)	17 (50.00)	0.0002	4 (8.69)	22 (47.82)	<0.0001	0.9950	0.8471
Fatigue	16 (47.09)	19 (55.88)	0.4684	19 (41.30)	27 (58.69)	0.0953	0.6058	0.8016
Psychomotor agitation	3 (8.82)	21 (61.76)	<0.0001	4 (8.69)	23 (50.00)	<0.0001	0.9838	0.2959
Aggressiveness	4 (11.76)	14 (41.17)	0.0060	5 (10.86)	18 (39.13)	0.0017	0.8998	0.8539
Anxiety	2 (5.94)	16 (47.05)	0.0002	2 (4.34)	23 (50.00)	<0.0001	0.7459	0.7941
Headache	5 (14.70)	10 (29.41)	0.1435	3 (6.52)	18 (39.13)	0.0002	0.2279	0.3676
Hair loss	4 (11.76)	6 (17.64)	0.4936	16 (34.78)	19 (19.56)	0.1008	0.0187	0.8278
Skin reactions	-	6 (17.64)	No Data	19 (19.56)	19 (19.56)	1.0000	No Data	0.8278
Diplopia or blurred vision	3 (8.82)	6 (17.64)	0.2831	12 (26.08)	15 (32.60)	0.4922	0.0505	0.1327
Dyspepsia	-	9 (26.51)	No Data	2 (4.34)	22 (47.82)	<0.0001	No Data	0.0531
Gingival hypertrophy	10 (29.41)	5 (14.70)	0.1435	-	10 (21.74)	No Data	No Data	0.4251
Tremor	3 (8.82)	11 (32.35)	0.0164	-	11 (23.91)	No Data	No Data	0.4033
Weight gain	2 (5.92)	8 (23.52)	0.0405	-	4 (8.69)	No Data	No Data	0.06662
Dizziness	4 (11.76)	9 (26.51)	0.1221	8 (17.39)	14 (30.43)	0.1426	0.4857	0.7018
Somnolence	19 (55.88)	20 (58.82)	0.8064	-	-	No Data	No Data	No Data
Memory impairment	7 (20.58)	12 (35.10)	0.1820	5 (10.86)	24 (52.17)	<0.0001	0.2286	0.1292
Sleep disturbance	6 (17.64)	8 (23.52)	0.5487	13 (28.26)	13 (28.26)	1.0000	0.2698	0.6338
Lack of concentration	1 (2.94)	4 (11.76)	0.1634	5 (10.86)	23 (50.00)	<0.0001	0.1835	0.0003

**Table 3 ijerph-19-04509-t003:** Clinical and statistical data of pediatric patients with epilepsy.

	Baseline		Followup	
	Monotherapy (*n* = 34)	Polytherapy (*n* = 46)	Monotherapy (*n* = 34)	Polytherapy (*n* = 46)
**Average number of ADRs per month (±SD)**	5.02 ± 0.62	5.12 ± 1.77	6.40 ± 1.65	8.78 ± 0.93 *
	*p* = 0.7536		*p* < 0.0001	
**Average seizure frequency per month (±SD)**	32.22 ± 12.21	21.12 ± 7.31	27.47 ± 12.45	15.98 ± 5.19 *#
	*p* < 0.0001		*p* < 0.0001	
**Average number of hospitalization days per month**	3.19 ± 0.80	2.55 ± 0.28 *	2.79 ± 1.02	2.09 ± 0.71 *
	*p* < 0.0001		*p* = 0.0005	
**Average number of outpatients neurologist visits per month**	2.01 ± 0.29	1.59 ± 0.31	1.21 ± 0.78 **	1.77 ± 0.51 *
	*p* < 0.0001		*p* = 0.0002	

SD—standard deviation; AE—adverse events; * statistically significant difference (*p* < 0.0001) vs. baseline in polytherapy; # statistically significant difference (*p* = 0002) vs. baseline in polytherapy; ** statistically significant difference (*p* < 0.0001) vs. baseline in monotherapy.

**Table 4 ijerph-19-04509-t004:** Comparative analysis of the pharmacotherapy regimen efficacy among pediatric patients with epilepsy.

Treatment Groups	Average SF ± SD [Med./95% CIs] in Baseline (*n* = 34)	Average SF ± SD [Med./95% CIs] in Followup (*n* = 46)	Average SFR ± SD [95% CIs]	Average % Decrease (Increase) in Seizure Frequency in Followup vs. Baseline	*p* Value
**Monotherapy**	32.22 ± 12.21 [15.14, 40.85]	27.47 ± 12.45 [11.48, 18.53]	0.85 [10.62, 31.38]	15	0.0930
**Polytherapy**	21.12 ± 7.31 [24.84, 48.24]	15.98 ± 5.19 [14.14, 21.96]	0.76 [22.59, 36.66]	24	0.0004

SF—seizure frequency; SFR—seizure frequency ratio; NS—no significance; SD—standard deviation; AEDs—antiepileptic drugs; CIs- confidence intervals.

## Data Availability

Not applicable.

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
