# Peer review of "Pharmacovigilance in Pediatric Patients with Epilepsy Using Antiepileptic Drugs"

_ijerph, 2022, doi:10.3390/ijerph19084509_

Round 1

Reviewer 1 Report

The goal of the study is to investigate the occurrence of adverse effects of antiepileptic drugs in pediatric epileptic patients on mono- or polytherapy. This study found that children on polytherapy were significantly more likely to develop AE compared to those on monotherapy, but a statistically significant improvement in seizure frequency was also observed in the group of patients on polytherapy.

Major comments:

  1. Are the monotherapy subjects with more severe epilepsy? Please add more description in Method.
  2. Comparison of monotherapy and polypharmacy with baseline and follow-up may be added to this study.
  3. Whether there is an adjustment for the severity of epilepsy of the subjects themselves, in the statistical analysis of this study (Table 2-4).

Minor comments:

  1. The full SFR should appear in the first introduction.
  2. Table2-4: NS – no significance can still show p-value.

Author Response

Major comments:

  1. Are the monotherapy subjects with more severe epilepsy? Please add more description in Method.

Response: If we consider the severity of epilepsy by the number of epileptic seizures, then yes - patients on monotherapy have more severe type of epilepsy, because they experienced much more seizures than patients under polytherapy. These data are presented in Table 3.

This information was not included in the methodology as we did not know it when developing the project of the study. Only the analysis of the patients' data made it possible to determine the frequency of seizures, which was described in the results section.

Nevertheless, information has also been added to Table 1.

  1. Comparison of monotherapy and polypharmacy with baseline and follow-up may be added to this study.

Response: The data were compared. This was shown in Table 2.

  1. Whether there is an adjustment for the severity of epilepsy of the subjects themselves, in the statistical analysis of this study (Table 2-4).

Response: The analysis of the parameters related to the disease stage is presented in Table 3.

Minor comments:

4.. The full SFR should appear in the first introduction.

Response: The abbreviation SFR first appeared in the text in the methodology section ( page 2, last sentence of the methodology). Nevertheless, as you suggest, an expansion of the abbreviation has also been added to the statistical analysis section.

  1. Table2-4: NS – no significance can still show p-value.

Response: thank you for your observations. Corrected, as you suggested.

Reviewer 2 Report

The authors investigated adverse drug reactions (ADRs) associated with antiepileptic drugs (AEDs) in pediatric population with epilepsy. The results show that more patients with polytherapy were affected with ADR than monotherapy patients.

Here are some comments:

  • 1) The actual numbers of patients should be included in the abstract.
  • 2) It seems that the term AE and ADR are used in the same manner. It would be useful to explain or use only one.
  • 3) The higher risk of polytherapy than monotherapy has been well-known since 1980s, and the present study is consistent with the past studies. I wonder what are the new findings and viewpoints of the present study given to clinical knowledge or the research field.
  • 4) A few minor correction should be needed, i.g., ‘Eighty epileptic child’ in Results section.

Author Response

1) The actual numbers of patients should be included in the abstract.

Response: as you suggested, the number of patients was added to the abstract.

2) It seems that the term AE and ADR are used in the same manner. It would be useful to explain or use only one.

Response: corrected, as you suggest.

3) The higher risk of polytherapy than monotherapy has been well-known since 1980s, and the present study is consistent with the past studies. I wonder what are the new findings and viewpoints of the present study given to clinical knowledge or the research field,

Response: Considering the differences in the availability of treatment, the differences in the approach to therapeutic regimens, as well as differences depending on the country, we believe that the analysis of the therapeutic regimen (monotherapy vs polytherapy) in terms of its effectiveness and the frequency of ARDs in Poland is still a topic worth taking scientific action and actualization.

4) A few minor correction should be needed, i.g., ‘Eighty epileptic child’ in Results section.

Response: corrected

Reviewer 3 Report

Kopciuch et al compared the side effects of polytherapy and monotherapy used for epilepsy treatment in this study. Some concerns and questions are shown below.

1. What dose AE mean in the abstract and throughout  the paper? antiepileptic ? adverse effects ? it is better not to use this abbreviation.

2. How many patients were included ?  from which hospital ?

3. A table including the information of age, gender, number of drugs used, treatment duration is suggested.

4. Line 67: what is the age range ? for young babies, how did they complete the questionnaire ?also the children may not give a reliable answer for those professional questions. One-time questionnaire-based were reliable.

5. Whether the age factor is associated with the adverse effects ?

6. No clear identification of the monotherapy and polytherapy. Dose polytherapy meas multidrugs used ? whether the number of drug used together is related to the number of adverse effects ? Details are needed for classify the monotherapy and polytherapy.

7. No indication of the name of the drugs used for epilepsy treatment.

8. What is the specific antiepileptic drugs used for each individual patients ? They were treated by same drugs or not ?

9. If they used different drugs for epilepsy treatment,  it is impossible to make comparison of different drug side effects ?

Author Response

  1. What dose AE mean in the abstract and throughout the paper? antiepileptic ? adverse effects ? it is better not to use this abbreviation.

Response: AE means adverse effect. It was removed/corrected, as you suggested.

  1. How many patients were included ?

Response: 80 children. The information was added in the abstract.

 from which hospital ?

Response: due to the blinded status of the reviewing process, it was not possible to provide full details of the hospital and department.

  1. A table including the information of age, gender, number of drugs used, treatment duration is suggested.

Response: These data are included in Table 1. Only gender information has been added, as you suggested.

  1. Line 67: what is the age range ? for young babies, how did they complete the questionnaire ?also the children may not give a reliable answer for those professional questions. One-time questionnaire-based were reliable.

Response: The age ranged from 2 to ≤ 18 years (based on the patient’s medical card).  The information contained in the questionnaires was obtained from children and/or their caregivers.

Information were added, as you suggest.

  1. Whether the age factor is associated with the adverse effects ?

Response: it was not assessed.

  1. No clear identification of the monotherapy and polytherapy. Dose polytherapy meas multidrugs used ? whether the number of drug used together is related to the number of adverse effects ? Details are needed for classify the monotherapy and polytherapy.

Response: Yes, polytherapy meas multi drugs used. We have not assessed whether the number of drugs used in polytherapy influences the incidence of ADRs. Thank you for your suggestion, we will certainly take it into account in the construction of the next study. However, additional information has been added in Table 1 to clarify the pharmacotherapy regimen.

  1. No indication of the name of the drugs used for epilepsy treatment.

Response: Additional informations has been added in Table 1 to clarify the pharmacotherapy regimen.

  1. What is the specific antiepileptic drugs used for each individual patients ? They were treated by same drugs or not ?

Response: some informations about medicines were added, as you suggest.

  1. If they used different drugs for epilepsy treatment, it is impossible to make comparison of different drug side effects ?

Response: The purpose of the study was to assess whether more ADRs were caused by a monotherapy regimen and polytherapy (and its effectiveness based on seizures frequency), not to analyze ADRs caused by individual drugs.

But we agree that it would be interesting to re-analyze it and we will certainly consider it in further research.

Reviewer 4 Report

The Authors of the manuscript raise a very important topic regarding the safety of pharmacotherapy in children. As we know perfectly well, the problem of the safety of using drugs in children results, among other things, from the fact that clinical trials are not conducted with their participation, hence the reaction to a given drug is often unpredictable. Moreover, metabolism in young people may differ from that in adults, so selecting an appropriate dose is also difficult. However, the work has many inaccuracies. Below is a list of considerations.

Introduction – antiepileptics are a large and diverse group of drugs. Diversified in terms of the mechanism of action as well as side effects. Throughout the very brief chapter, there aren't even mentioned specific possible side effects.

It is obvious that side effects may appear, but the term: "Adverse effects of AEDs are normal" - is not very accurate, especially that not all of them occur and not in every person. Besides, since they are called side effects, that is, they should not be the norm, although they may occur during the application of the therapeutic dose.

Method

What questions were asked of the patients? A sample questionnaire should be translated into English and attached as additional material.

What medications were used? There is no information at all on this subject throughout the work. Lines 78-79 is not enough. In addition, possible side effects that may occur during treatment with a given drug should appear at work.

Many anti-epileptic drugs will have similar side effects, but the frequency will vary. So it cannot only be written about a side effect without reference to therapy.

It is known that side effects are correlated with the concentration of the drug in the body. Stabilization of the concentration, e.g. after using carbamazepine, is 5-6 weeks. Why, then, were such timing of the study chosen? Why was the second study conducted after 3 months? There is no justification for this choice at work

Did the patients make a record of the occurrence of a given side effect so as to precisely determine its frequency?

Was the first examination carried out regardless of the length of treatment with a given drug? The term "stable dose of AED for at least three months" means that for some patients it will be a year and for others it will be exactly 3 months. During this time, the number of side effects for both people, even using the same drug, will be different (stabilization of the concentration).

Tables - does the number “n” refer to people treated with monotherapy and polytherapy? Or does it refer to baseline and follow-up definition (Table 2 and 3). Or are the overlapping numbers a coincidence?

Line 154 - wrong write

Author Response

Method

  1. What questions were asked of the patients? A sample questionnaire should be translated into English and attached as additional material.

Response: The questionnaire was added as supplementary material as you suggested. Besides, all questions about ADRs was included in the Table 2.

  1. What medications were used? There is no information at all on this subject throughout the work. Lines 78-79 is not enough. In addition, possible side effects that may occur during treatment with a given drug should appear at work.

Response: Additional informations were included in result section and Table 1.

  1. Many anti-epileptic drugs will have similar side effects, but the frequency will vary. So it cannot only be written about a side effect without reference to therapy.

Response: Thanks for your valued opinion. We agree with you. However, the aim of the study was not to find out which drug causes the most ADRs and what, but to analyze whether monotherapy or polytherapy would be more effective and safe. Additional informations about pharamacotherapy regimen were included in the Table 1.

  1. It is known that side effects are correlated with the concentration of the drug in the body. Stabilization of the concentration, e.g. after using carbamazepine, is 5-6 weeks. Why, then, were such timing of the study chosen? Why was the second study conducted after 3 months? There is no justification for this choice at work

Response:  Thanks for your valued opinion. We agree with you.  However, the patients participating in the study were our long-term patients. On the other hand, the chosen time horizon of the study was necessary to meet the given requirements related to the implementation of the study.

Thank you again for your opinions. We will certainly consider it when developing another study in this field.

  1. Did the patients make a record of the occurrence of a given side effect so as to precisely determine its frequency?

Response: Yes, patients were asked about the frequency of occurrence of ADRs per month in the last 3 months. These data are presented in Table 3.

  1. Was the first examination carried out regardless of the length of treatment with a given drug? The term "stable dose of AED for at least three months" means that for some patients it will be a year and for others it will be exactly 3 months. During this time, the number of side effects for both people, even using the same drug, will be different (stabilization of the concentration).

Response: Thanks for your valued opinion. Unfortunately, we did not take this into account.  We agree with you, and certainly we will consider it when developing another study in this field.

  1. Tables - does the number “n” refer to people treated with monotherapy and polytherapy? Or does it refer to baseline and follow-up definition (Table 2 and 3). Or are the overlapping numbers a coincidence?

Response: Thank you for your remark. It was a mistake. Corrected according to the suggestion

  1. Line 154 - wrong write

Response: Corrected, as you sugested.

Round 2

Reviewer 1 Report

In the revised manuscript, NS still does not show the p-value in Table2-4. Please correct the manuscript for Table2-4.

Author Response

In the revised manuscript, NS still does not show the p-value in Table2-4. Please correct the manuscript for Table2-4.

Response: Sorry, I misunderstood comment you've made before. The p values were added as you suggest this time.

Reviewer 3 Report

No more concerns.

Author Response

No more concerns

Response: Thank you

Reviewer 4 Report

The Authors responded to almost all of the comments. However, the "±" symbol (line 157) is still missing.

Author Response

The Authors responded to almost all of the comments. However, the "±" symbol (line 157) is still missing.

Response: Corrected
